# Linguistic metaconcepts can improve grammatical understanding in L1 education evidence from a Dutch quasi-experimental study

**Jimmy van Rijt** [1,2,3] *, **Debra Myhill** [3], **Sven De Maeyer** [4], **Peter-Arno Coppen** [5]

**1** Tilburg Center of the Learning Sciences, Tilburg University, Tilburg, The Netherlands, **2** Fontys University of Applied Sciences, Sittard, The Netherlands, **3** Graduate School of Education, University of Exeter, Exeter, United Kingdom, **4** Department of Educational Sciences, University of Antwerp, Antwerp, Belgium, **5** Radboud Teachers Academy, Radboud University, Nijmegen, The Netherlands

* j.h.m.vanrijt@tilburguniversity.edu

## Abstract

This mixed-method quasi-experimental study examined whether metaconceptual grammar teaching impacts on (a) students' L1 grammatical understanding, (b) their 'blind' use of grammatical concepts and (c) their preference of using explicit grammatical concepts over everyday concepts in explaining grammatical problems. Previous research, involving single group pre-postintervention designs, found positive effects for metaconceptual interventions on secondary school students' grammatical reasoning ability, although a negative side effect seemed to be that some students started using grammatical concepts 'blindly' (i.e., in an inaccurate way). While there are thus important clues that metaconceptual grammar teaching may lead to increased grammatical understanding, there is a great need for more robust empirical research. The current study, involving 196 Dutch 14-year old pre-university students, is a methodological improvement of previous work, adopting a switching replications design. Bayesian multivariate analyses indicate medium to large effects from the metaconceptual intervention on students' grammatical understanding. The study found a similar effect of the intervention on students' ability to use explicit grammatical concepts over everyday concepts in tackling grammatical problems. No evidence for increased 'blind' concept use as a negative byproduct of the intervention was found. Additional qualitative analyses of in-intervention tasks provided further evidence for the effectiveness of metaconceptual interventions, and seemed to indicate that cases of blind concept use, rather than being a negative side effect, might actually be part of a gradual process of students' growing understanding of grammatical (meta)concepts. We discuss these findings in relation to previous work and conclude that linguistic metaconcepts can improve L1 grammatical understanding.

## 1. Introduction

*Understanding* can be seen as one of the primary goals of education. According to Baumberger, Beisbart and Brun [1] it is crucial to think about what understanding entails, because it

**Data Availability Statement:** All relevant data are within the paper and its Supporting Information files. Data can also be found on the OSF repository

at: DOI 10.17605/OSF.IO/EGKJD (https://osf.io/egkjd/?view_only=None).

**Funding:** YES The research from this article was funded by the Netherlands Organisation of Scientific Research (NWO) under grant number 023.009.034, awarded to Jimmy H.M. van Rijt.

**Competing interests:** The authors have declared that no competing interests exist.

appears to be 'a central good that we try to realize when we think about the world', which surpasses knowledge as such [2, 3]. In addition, understanding is a central goal in the sciences [1], which makes it only natural that education should focus on understanding. But what does it mean to understand something? It might be argued that understanding something can be measured by the ability to explain it. Furthermore, while there appears to be a growing awareness that understanding cannot be identified as or reduced to explaining, it is clear that there are strong conceptual links between them [4].

What is becoming more and more accepted is the idea that understanding comes in degrees, i.e., one can have partial understanding of a certain phenomenon rather than a complete understanding [4]. Among other things, understanding things entails accounting for supposed facts (*factivity condition*, [5]), and being able to apply that understanding 'to actual and counterfactual cases by inferring conclusions and giving explanations' (*ability condition* [6, 7]). In this process, it is important to note that understanding requires more than simply reproducing isolated pieces of knowledge. Rather, anyone who wants to understand something should be able to see how different pieces of knowledge are connected [8]. Understanding one thing should therefore lead to the ability to understand similar cases [1]. This paper focuses particularly on the notion of *grammatical understanding* in an L1 context, which Macken-Horarik et al. [9] define as 'any grammatically informed knowledge about language'. The paper deals with the question of how grammatical understanding can be improved by means of explicit instruction, and it takes the Dutch secondary school context as its frame of reference, although its relevance is much broader.

## 1.1 Grammatical understanding

Strengthening students' grammatical or metalinguistic understanding is one of the most challenging issues for L1 language teachers all over the world [10–14]. At the same time, the question of how such understanding can be achieved is underresearched. It has been pointed out by several authors that most of the research papers dealing with grammar teaching or learning in L1 contexts have contributed to debating rationales for grammar teaching, rather than attempting to establish how grammar is best taught, and how students can best develop a good conceptual understanding of the subject matter [12, 15–18]. This is particularly problematic because grammar has made a strong comeback in L1 language education in many educational jurisdictions [19–21], or it has always been prominently present, as is the case in the Netherlands [22, 23]. This means that many teachers either want to teach grammar, or that they are forced to teach it. And while many teachers consider grammar teaching useful or valuable, particularly in the Dutch context [22, 24], they also struggle with how to teach it effectively. Several studies have identified limitations in teachers' or student teachers' grammatical subject knowledge [25–29]. In fact, due to limitations in grammatical knowledge, many teachers experience feelings of anxiety when teaching grammar [30–32]. This makes it especially challenging for them to achieve a good conceptual understanding in their pupils, although even grammatically skilled teachers tend to struggle with issues of effective pedagogy. In the Netherlands, for example, where grammar is being taught a lot, especially in the lower classes of secondary education (see [23]), rules of thumb and audit questions meant to identify parts of speech in isolated sentences reign supreme, especially in commonly used text books, and there are only few viable pedagogical alternatives [28].

Internationally, there are two areas where grammatical understanding plays a key role. The first area deals with grammatical understanding within the context of writing education, where grammatical knowledge is used to strengthen students' writing skills [33]. In such cases, grammatical knowledge (or metalinguistic knowledge more broadly) is seen as a means to an

end. It has been well established by now that traditional grammar teaching, in which isolated sentences are typically parsed by means of superficial tricks [34, 35], does not contribute to writing development at all [36]. This type of grammar teaching, 'focusing more on the mechanic articulation of a rule without any corresponding understanding of its grammatical implications' [12] is dominant in the Netherlands [23], as well as in many other educational contexts throughout the world [17, 37]. On the other hand, less traditional and more contextualized forms of grammar teaching, such as those informed by Systemic Functional Linguistics (SFL, cf [38], in which grammar is actively addressed in the context of writing, do have a powerful impact on writing ability [21, 39, 40] and on talking about texts in a more general sense [41].

The second area where grammatical understanding is crucial is in the learning of grammar as a system. From this perspective, grammar teaching is a valuable goal in itself, and the goal of grammar teaching in such settings is not to enhance writing, but to develop an understanding of how language works [42, 43]. Hulshof [44] attributes *cultural* value to this perspective, and he distinguishes it from the grammar-writing perspective which he classifies as being *instrumental*. Myhill [15] makes a similar distinction, namely between seeing grammar predominantly as *useful* (related to writing proficiency) versus *valuable* (related to understanding grammar as a system). The importance of gaining grammatical understanding from a cultural point of view is underlined by several researchers, and it is incorporated into the curricula of many countries around the globe in some shape or form [15, 16, 18, 45, 46]. Moreover, a good understanding of grammar as a system is likely to be a necessary condition for the teaching of grammar for writing [47]. Studies on grammatical understanding are therefore likely to impact both areas of grammar teaching [17]. The current paper, however, will focus predominantly on the second area of grammar teaching (learning grammar as a system).

## 1.2 Enhancing grammatical understanding

According to Myhill [12], there are three reasons why learning metalinguistic knowledge can be problematic. Firstly, due to learners' previously acquired misconceptions, which are frequently created by teachers and textbooks; secondly, by specific grammatical characteristics; and thirdly, because of cognitive difficulties related to the conceptual demands of grammar. It has been suggested that the latter two problems can be diminished by enriching traditional grammatical knowledge with linguistic metaconcepts [48], since modern linguistics has generated many insights and (meta)concepts that could be used to better understand traditional grammatical terminology [43, 49]. For example, traditional grammatical concepts such as direct and indirect objects are often misunderstood by students because they lack an understanding of the metaconcept of *valency*, (i.e., the grammatical and semantic selection of arguments by verbs) which underlies those concepts [50]. For example, different verbs have different slots available for subjects and objects: the verb *to shiver* only has a slot available for a subject (e.g., 'HE shivers', not 'HE shivers SOMETHING'), whereas the verb *to read* has a slot available for both a subject and a direct object (e.g., SHE reads SOMETHING). Some verbs, such as the verb *to give*, even have three slots, namely for a subject, a direct and an indirect object (e.g., THE WIZARD gave THE CAT SOME FOOD). Valency can therefore be called a metaconcept, because it conceptually overarches other concepts (subject, direct/indirect object). While the metaconcept of valency is absent from traditional school grammar, it is central in modern linguistic theory [48, 50]. Following Van Rijt [51], *linguistic metaconcepts* can thus be defined as 'higher-order concepts that facilitate the understanding or categorization of the lower-order concepts they thematically organize'. Subsequently, higher-order concepts such as *valency* can be used to understand lower-order concepts such as *direct objects*, but not

the other way around. Approaches in which more general metaconcepts are aimed at first (*valency*) before refining that understanding with related subordinate concepts (*subject*, *objects*) have been found pedagogically useful in other fields, such as in biology education [52] or history education [53], as they are aimed at understanding larger parts of a system before looking at the components of which the system comprises. This allows students to see more clearly how different concepts are organized [54]. However, research into metaconceptual approaches in L1 grammar learning has been quite scarce. We will discuss the only research that–to the best of our knowledge–addresses the role of linguistic metaconcepts in L1 grammar education.

While being limited in number, previous exploratory studies have revealed that interventions targeting linguistic metaconcepts can have a significant impact on grammatical reasoning ability, which is likely to reflect grammatical understanding. Van Rijt, De Swart, Wijnands and Coppen [55] demonstrated that an intervention in which conceptual connections are made between concepts from traditional school grammar and metaconcepts from linguistic theory can significantly improve university students' grammatical reasoning ability (cohen's $d$ = 0.62). Moreover, a multilevel regression analysis revealed that students who used underlying linguistic metaconcepts explicitly in tackling unknown grammatical problems, especially when they were related to traditional grammatical concepts, produced better grammatical reasonings than students who did not use underlying metaconcepts. Describing concepts in their own words in their grammatical reasoning (*implicit* concept use) was not positively rated by linguistic experts. Experts thus seem to favor students' use of explicit grammatical concepts in their reasoning over implicit, everyday concept use.

Van Rijt, Wijnands and Coppen [56] investigated to what extent these findings could be transferred to 14-year old pre-university students. To this end, they designed an intervention in which four linguistic metaconcepts (*predication*, *valency*, *complementation* and *modification*) were related to concepts from traditional school grammar (*subject*, *objects*, etc.), and in the pre- and post-test, they encouraged students ($N$ = 119) to reason about unknown grammatical problems. Students' reasonings were then rated by means of comparative judgment [57, 58].

Van Rijt et al.'s [56] intervention was underpinned by five design principles to maximize grammatical understanding. The same design principles have been adopted in the current study (cf. Table 1).

In analyzing the students' reasoning about unknown grammatical problems, Van Rijt et al. [56] found that students' grammatical reasoning improved significantly (Cohen's $d$ = 0.46) on the target items (items which could be tackled either with concepts from traditional grammar, linguistic metaconcepts or a combination of both), whereas their reasoning did not improve on the filler items (items which could not be tackled with the aforementioned concepts). This indicates that it is unlikely that the students' improvement could be attributed to a testing effect [67].

Although students benefitted from the Van Rijt et al. [56] intervention overall in terms of their grammatical reasoning ability, some of them showed incomplete acquisition of the linguistic metaconcepts. This means that they used these concepts in their reasoning 'blindly', i.e., in an inappropriate way, or just for purposes of 'name dropping' instead of for demonstrating understanding. This appeared to be a negative side effect of the intervention.

## 1.3 The current study

While the Van Rijt et al. [56] study provided empirical evidence to show how secondary school students can benefit from interventions based on linguistic metaconcepts for principled

**Table 1. Explanation of the design principles from the Van Rijt et al. [56] study, adopted in the current study.**

| Design principle | Clarification and grounding in literature |
|---|---|
| (1) Relate underlying metaconcept(s) to concepts from traditional school grammar | Establish active connections between metaconcepts and the traditional concepts, in such a way that the teachers first aim to establish an understanding of the metaconcept before moving on to specify that metaconcept with traditional concepts (e.g., first establishing an understanding of *valency* before establishing understanding of *objects*). [48, 52] |
| (2) Use inductive exercises to introduce linguistic (meta)concepts | Stimulate students' own language intuitions about a metaconcept by means of inductive exercises [35, 59]. |
| (3) Expose students to limited degrees of grammatical uncertainty | Clear-cut answers hardly exist in grammatical analysis, making grammar an ill-structured knowledge domain [35, 60], for which reflective judgments are required [61]. This means that students will need to develop a reflective attitude towards grammar that facilitates dealing with grammatical uncertainty [60]. This has been done by exposing students to limited degrees of grammatical uncertainty (e.g., by using odd one-out tasks), so they would use their conceptual knowledge to take different perspectives into account and make informed decisions about grammatical problems that could be viewed from multiple angles. Such odd one-out tasks have been proven effective in learning about grammar [29, 62]. |
| (4) Stimulate talk | The intervention stimulated various forms of talk (e.g., dialogic talk [26] or forms of exploratory talk [63], which has been proven effective in the learning and teaching of grammar [14, 18, 64]. |
| (5) Support teachers' scaffolding strategies | Teachers were encouraged to guide students in evaluating possible solutions to a grammatical problem, rather than simply 'giving the right answers'. They were instructed to pay particular attention to guiding the use of conceptual terminology, as was found to be crucial in teacher scaffolding in guiding historical reasoning in history classes [65, 66]. |

grammatical understanding, it was exploratory in nature. For more robust conclusions, the methodological design needs to be improved (in order to rule out test effects more clearly, for example [68]), and the total number of observations needs to be increased. The current study has done so. In addition, the study examined the potential role of blind (meta)concept use as a negative byproduct of metalinguistic interventions. It also investigated whether metaconceptual interventions affect students' preference of linguistic concepts over their everyday explanations of these concepts, since the former is being much more valued in grammar classrooms than the latter, as was demonstrated by Van Rijt, De Swart, Wijnands and Coppen [55]. In addition, while Van Rijt et al. [55, 56] focused on the effect of a metaconceptual intervention on reasoning quality, there is also a need to investigate the effect of such interventions on grammatical *understanding* specifically (rather than reasoning), since written reasoning tasks rely heavily on students' general writing abilities, which can negatively influence their linguistic reasoning ability, as many students experience difficulties in writing [36].

Because of the strong conceptual links between understanding and the ability to explain something [4], students' explanations of grammatical issues are of central interest to the present paper.

## 1.4 Research questions

The current study aims to provide more robust empirical evidence of the potential efficacy of a metaconceptual approach to L1 grammar learning. More specifically, it aims to answer the following research questions:

1. What is the effect of a metaconceptual intervention on students' grammatical understanding, and to what extent can their progress be attributed to a testing effect?

2. To what extent is blind concept use (indicative of poor understanding) a negative byproduct of the intervention?

3. To what extent does the intervention encourage students to favor explanations in which relevant grammatical concepts are used (explicit concept use) over explanations in which they are not (implicit concept use)?

4. What do students' responses to the intervention tasks reveal about their grammatical understanding?

## 2. Method

### 2.1 Research design

The current study deployed a quasi-experimental research design with switching replications. A switching replications design controls most threats to internal validity, being well suited to rule out the possibility of a testing effect. It has several other advantages, such as a delayed posttest for the experimental condition, the ethical benefit of both groups receiving the same treatment [67] and the fact that the main intervention effect can be repeated, which is very much appreciated in light of the limited reproducibility of educational research [69]. Fig 1 gives a general overview of the switching replications design.

In the business as usual time frame, teachers taught their regular schedule (e.g., reading, writing, literature), but they were not allowed to teach any grammar to ensure that any effects at the various occasions of measurement were not confounded by students who had received extra grammar. The time frame of two weeks between each measurement was chosen because the teachers only had time for a limited intervention that could take up no more than four lessons (see section 2.2). It should be noted that the current design is not intended to compare a traditional approach to grammar (i.e., traditional parsing) to a metaconceptual approach; rather, the study focuses on the efficacy of a metaconceptual approach in itself. See De Maeyer [70] for more details.

### 2.2 Participants

Seven teachers from five different secondary schools in the Netherlands voluntarily participated in this study. Depending on their schedule, they were allocated to either group 1 (experimental) or to group 2 (control). One of the teachers was male. On average, teachers had 8.6 years of teaching experience ($SD$ = 4.3). Their experience did not differ significantly over both conditions.

A total of 196 ninth grade pre-university students participated in this study (mean age = 14.03 years, $SD$ = 0.45), 90 of whom were male (45.9%). Fisher's exact test indicated no

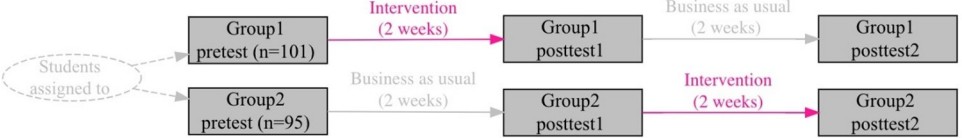

**Fig 1. General overview of switching replications design.** Group 2 thus serves as a control to group 1. When the control group later receives the treatment, the original treatment group then serves as a continued-treatment control [67].

significant differences between the distribution of male and female students across the conditions, nor were there significant age differences between the students across both conditions.

**2.2.1 Ethics statement.** Both the teachers and students involved gave active written consent for using the anonymized data for scientific research, as well as the students' parents or guardians. One parent/guardian withheld consent, so their child's data was removed from the data file. Schools and participating teachers also approved of the investigation in writing. The study was approved by the ethical committee of the Radboud University Nijmegen, under case number 2019–9383.

## 2.3 Intervention

Since the design principles from Van Rijt, Wijnands and Coppen [56] were found to be effective in enhancing students' grammatical reasoning ability, these same design principles were adopted (see Table 1). The intervention used similar assignments and was of similar length as the one in the Van Rijt et al. study (4 lessons of 50 minutes, the standard time for Dutch secondary school lessons) [56]. Based on the feedback of the teachers who were enrolled in the Van Rijt et al. [56] study, some improvements to the intervention were made. The most important one was that rather than focusing on four different (related) linguistic metaconcepts, the intervention focused on just one, namely *valency*. This way, students would have the opportunity to process the newly taught information more effectively, limiting the amount of new linguistic terminology. Similar to the Van Rijt et al. [56] study, the intervention was presented as an enriched form of repeating the knowledge students were already meant to possess. All students had received years of traditional grammar teaching, so they were familiar with some of the traditional grammatical concepts from the intervention to some degree (e.g., verb, subject, direct object).

For the intervention a student assignment booklet was developed, with several pen-and-paper assignments per lesson. Three of the assignments, in which students focused on grammatical reasoning, revolved around group work and were performed outside of the booklet. Other assignments that relied on group or pair work could be written down within the booklet individually. Detailed information on the intervention is given in Appendix A in S1 File, to strengthen the study's replicability, following the recommendation of Rijlaarsdam et al. [71].

To facilitate teachers as much as possible, they were given an elaborate instruction manual, containing a background to the study and its underpinning design principles, as well as answers to the assignments and things to look out for. Teachers were told that they could deviate from the specifics of the intervention to a limited degree if the situation called for it, as long as they would act in the spirit of the design principles and report any changes they had made to the assignments. This was done to maximize ecological validity [21, 72]. Teachers did not report any major deviations from the intervention.

## 2.4 Instruments

For the present study, we developed a Test for Grammatical Understanding (TGU). The TGU consisted of twelve multiple-choice questions, in which a question was asked related to the concepts from the intervention (see Table 2 for an example, and Appendix B in S1 File for the full versions of the TGU). Three similar versions of the TGU were developed to suit the needs of the research design (i.e., three moments of measurement), and the three TGU versions were counterbalanced to avoid any unintended order effects [67], meaning that students were randomly assigned with one of the three versions at each moment of measurement. Crucially, the questions from the TGU did not contain any explicit references to the metaconcept of valency to ensure that students would have a fair chance of tackling these questions even before the

**Table 2. Example of a question (see [29]) with the different multiple-choice options (translated from Dutch by the authors).**

| | Question: Why is the sentence 'My grandfather always smokes a lamppost' ungrammatical? | | |
|---|---|---|---|
| | **Alternatives** | **Type** | **Score (points)** |
| A | The verb 'to smoke' imposes restrictions on the meaning of the direct object | Maximum understanding | 2 |
| B | 'A lamppost' cannot be the direct object of the verb to smoke | Partial understanding | 1 |
| C | The verb 'to smoke' selects a mandatory direct object in normal sentences | Blind concept use | 0 |
| D | In a normal world it is hard to imagine that a lamppost is smoked | No concept use | 0 |

In the TGU, questions and answers were randomized.

intervention was enrolled. Instead of asking students about valency directly, the questions were designed so that they could be dealt with more easily if the metaconcept of valency and the related traditional concepts were properly understood.

Each of the twelve questions was then provided with four multiple-choice alternatives, and students were instructed to choose *the best alternative*, i.e., the alternative that they felt was the best explanation of the case at hand. The development of these alternatives was closely informed by the literature on understanding, as well as by some of the findings from Van Rijt et al. [56]. This means that all questions were provided with four (randomly ordered) answers. First, an answer that conveyed the core insight that was aimed at, which constituted the level of *maximal understanding*. Second, an answer that conveyed *a part* of the appropriate insight, to do justice to the fact that understanding comes in degrees, as Baumberger [4] suggests. Third, one of the answers contained a concept that was used *blindly*, meaning that the answer contained a term that was irrelevant for answering the question at hand (i.e., a case of name dropping as was found in Van Rijt et al. [56] to test whether students would be tempted to choose for a difficult term rather than for the appropriate insight). And finally, one of the answers used *no grammatical terminology* whatsoever, but instead represented an answer based on everyday language and experiences.

Students could score different points depending on the answers they chose. 2 points could be scored for choosing the correct insights (maximal understanding); 1 point for choosing a part of the correct insight. And finally, since Van Rijt et al. [56] found no quality differences between *blind concept use* and *no concept use* in students' grammatical reasoning, these answers both received a score of 0 points if chosen.

Thus, we were interested in three core variables based on the TGU: *Understanding Score* (total number of points), *Blind Score* (*N* of times a student chose a blind alternative) and *No Concept Score* (*N* of times a student chose the alternative without any explicit grammatical concepts).

One version of the TGU was first pretested on 7 students training to become Dutch language teachers, who were enrolled in a Bachelor of Education Programme. A corrected version based on their feedback was then pretested once more in a Dutch secondary school class, involving 19 ninth grade pre-university students (mean age = 14.05; *SD* = 0.62), 11 of whom were female (57.9%). P-P plots were then created to check for normal distributions. This was done for the core variables of Understanding Score, Blind Score and No Concept Score.

No abnormalities were found for each of these variables. From this, we concluded that the first version of the TGU was a suitable instrument for measuring grammatical understanding, which was confirmed by multiple teachers and linguists.

We then developed two other TGU versions with questions derived from the first version. Because they were conceptually identical to the first version, these versions were not pretested separately anymore. A parallel study [29] found that the TGU can distinguish between more or less advanced grammar students (i.e., student teachers training to become Dutch language teachers outperform secondary school students on the test), which is indicative of the instrument's validity.

Initial analyses of the data (One Way ANOVA) indicated that there were no significant differences between the three versions of TGU on the Understanding Score variable at M1: $F(2, 182) = 2.167$, $p = .117$). Significant differences were found for the Blind Score variable ($F(2, 182) = 11.15$, $p = < .001$) as well as for the No Concept Score variable ($F(2, 182) = 16.20$, $p = < .001$). Additional Bonferroni adjusted post hoc tests indicated that for both variables, version 1 significantly differed from version 2, and version 2 from version 3. For this reason, we controlled for TGU version in the models on the variables Blind Score and No Concept Score.

### 2.5 Procedures

The TGU was administered by the principal investigator one or two days prior to the start of the intervention, depending on practical possibilities. In some cases, the teachers administered the TGU themselves, following close instruction from the researchers. After the intervention, the TGU was administered within one or two days. In all cases, measurements were taken two weeks apart. The TGU was administered during regular hours of Dutch language class, and students had 15 minutes to complete the TGU individually on paper.

### 2.6 Implementation fidelity

To ensure that teachers had completed the intervention as intended, we measured implementation fidelity [73] via teacher logs and an analysis of the booklets that the students were provided with. We did not engage in any classroom observations to avoid evoking potential observer effects [74, 75]. Instead, teachers were asked to report for each lesson whether any deviations from the intervention had occurred (either planned or accidental), and whether they had finished all of the assignments within the given time. They were also requested to indicate if any other issues had arisen that could possibly have influenced the students' learning. They were encouraged to give their honest assessment, as they were told that researchers were not there to judge them, but simply to gain an understanding of how the intervention was executed.

In analyzing the students' booklets, we first assessed how many of the assignments had been answered. In addition, we qualitatively investigated in some detail three key assignments at various stages of the intervention to gain a deeper sense of how the students had performed in the lessons.

## 3. Data analysis

### 3.1 Assumptions check and outlier analysis

To assess whether the quantitative data were normally distributed, a P-P plot was created for each dependent variable (Understanding Score, Blind Score, No Concept Score) at each moment of measurement (M1, M2, M3). None of the P-P plots indicated any violations of a normal distribution. In addition, we checked for any issues regarding linearity and homoscedasticity, using regression analyses to create scores on all variables with condition (experimental vs. control) as a factor, plotting standardized residuals against predicted values. The plots did not indicate any non-linearity or heteroscedasticity. In the regression analyses, we also

applied casewise diagnostics for detecting potential extreme outliers ($> 3$ $SD$ from the regression line). Across the three variables and measurements, no extreme outliers were found.

## 3.2 Missing data

Most of the 196 students had participated in all measurement moments, and all of them had participated in at least 2 measurement moments. At M1, 11 students' data could not be taken into account due to illness or absence (attrition: 5.6%). At M2, this was true for 6 students (attrition: 3.1%). At M3 the attrition rate was higher, due to the illness of one of the teachers, resulting in the loss of one classroom of data. At M3, 36 cases were missing (attrition: 18.4%). Since we had student data for two moments of measurement in each case, we did not delete missing cases from our data in the multilevel modeling (see 3.3.). Attrition under such circumstances is not problematic in multilevel analyses [76, 77] as MLA can confidently estimate parameters based on the other two values.

## 3.3 Quantitative analysis of intervention effects

Our first three research questions (i.e., what is the impact of the intervention on (1) students' grammatical understanding, on (2) students' blind concept use and on (3) students' explicit concept use) were explored quantitatively. The quantitative analyses were done for each of the dependent variables (Understanding Score, Blind Score, No Concept Score) separately. We approached the data with a multivariate model, taking the three scores for a variable ($M1_i$, $M2_i$ and $M3_i$) as three dependent variables in a single model. The model is formally written as in Eq 1.

$$\begin{bmatrix} M1_i \\ M2_i \\ M3_i \end{bmatrix} \sim MVNormal(\begin{bmatrix} \mu_1 \\ \mu_2 \\ \mu_3 \end{bmatrix}, \Sigma_1)$$

$$\mu_1 = \beta_1,$$

$$\mu_2 = \beta_2,$$

$$\mu_3 = \beta_3$$

In this model we assume that the scores at each moment of measurement are coming from a multivariate normal distribution with a specific average ($\mu_1$, $\mu_2$ and $\mu_3$) for each measurement moment and an error variance-covariance matrix ($\Sigma_1$) assuming the residuals to be correlated.

This basic model (Model 1) is extended stepwise to model the effect of the intervention. In Model 2, we add the effect of a dummy variable identifying which condition a pupil is assigned to (which has value 1 for pupils in Group 1 who receive the intervention between occasion 1 and occasion 2 and value 0 for pupils in Group 2) as a predictor for $\mu_2$. This model 2 assumes that pupils in Group 1 will score differently at measurement moment 2 because they received the intervention while the students in the other condition did not. At measurement moment 3 we no longer expect a difference between both conditions as we assume that pupils in Group 2 now received the intervention and the experimental group did not. Moreover, this model also assumes that pupils in Group 1 will still benefit from the intervention and score at a similar level than pupils who just received the intervention (pupils in Group 2).

Two additional models will be tested. In Model 3 we also include an effect of the dummy-variable Group 1 for $\mu_1$ to test whether both groups did not score differently at the start of the

intervention. To test the assumption that both groups score similar at the end of the experiment (when they both received the intervention) we include the effect of the dummy-variable Group 1 for $\mu_3$ in Model 4.

All models will be estimated within a Bayesian inference framework, allowing us to get a posterior probability distribution of the parameter values of interest [78, 79]. The analyses are implemented in the probabilistic programming language Stan [80] making use of R [81] and the package *brms* [82] as a starting point for the model definition in Stan. For the estimation we made use of flat priors as implemented in *brms*. We ran 6 chains of 6500 iterations with 1500 iterations of warm-up resulting in 30 000 samples of estimates.

For each dependent variable we will compare the four alternative models on their model fit, using leave-one-out cross-validation (loo). Cross-validation prevents overfitting as this method penalizes models with more parameters [78, 83]. We determined the out-of-sample predictive performance via Pareto smoothed importance-sampling [84] and estimated the loo as sum of the expected log predictive density (elpd). To compare the predictive quality of the four models we used the $\widehat{\Delta elpd}$ and the looic value.

In a final step, we elaborated on the model with the best fit in the previous step by adding the effects of gender (Model 5) to see whether boys and girls performed differently, as well as the version of TGU the students had been given (Model 6). The model fit of these two models are compared with the model fit of the model that fitted best in the previous step to determine the final model that will be used to make inferences on the parameters in the model.

For the inferences, we visualized the posterior probability distributions of the parameter estimates, reported the credible intervals (CI's) and highest density intervals (HDI) for the parameter estimates and calculated effect sizes (Cohen's *d*) based on these posterior distributions. While traditionally, effect sizes of 0.5 and 0.8 constitute medium and large effects, respectively, we followed the interpretation of effect sizes by Calin-Jageman and Cumming [85], who indicate that the average effect size in educational research is 0.4. They therefore assume that effect sizes of 0.2, 0.4 and 0.6 constitute small, medium and large effects. All data, R scripts and Stan scripts are available on Open Science Framework (https://osf.io/egkjd/?view_only=None).

## 3.4. Qualitative analysis of intervention assignments

**3.4.1 Key intervention assignments.** Three key assignments were identified from the intervention that were analyzed in more detail to investigate how the students had performed these tasks, and to examine their level of understanding. Key assignment 1 entailed that students needed to explain in their own words how it was possible that one sentence might contain direct or indirect objects, whereas another sentence might not (lesson 1, after having been inductively introduced to the metaconcept of valency). The second key assignment (lesson 2) was more communicatively oriented, and invited students to reflect on a multimodal humanitarian aid poster in which the valency was deliberately reduced. They were asked to indicate whether there was something strange about the valency in the poster, and they were encouraged to relate their observation to any rhetorical effects that such use of valency might have. Key assignment 3, finally, comprised of two odd one-out tasks (lesson 4), in which students were asked to indicate which of the three grammatical items was the odd one-out, using a prescribed format (X is the odd one-out, because the other two . . .), to ensure that students addressed both the oddness of the odd one-out and the similarity in the other options. See Appendix C in S1 File for more details on these key assignments.

**3.4.2 Qualitative analysis.** The key assignments have been analyzed qualitatively and inductively in NVivo, following the constant comparison method [86] adopting Wellington's

four recommended stages of analysis: *immersion*, *reflecting*, *taking apart* and *synthesizing* [87]. In the initial stage of immersion, the first two authors of the present paper independently read through the data, establishing tentative open codes. These initial codes were then compared several times to develop a shared understanding of the data. In this stage, the first two authors met on a regular basis, continuously refining the codes, both individually and later together. In the final stage of coding, codes were synthesized (axial coding) based on inter-relationships and conceptual similarity. An overview of axial codes and subcodes is provided in Tables 5–7.

## 4. Results

### 4.1 Implementation fidelity

The metaconceptual intervention appears to have been well implemented overall. Of the 196 booklets that were handed out, 177 were returned and could be traced back to the owner. On average, students completed 77.4% of the assignments from the booklet, which is a percentage nearing the 80% limit that studies on effective teaching maintain [88]. In addition, all of the teachers had managed to complete 2 out of 3 tasks related to group work outside of the booklet; two of them had managed to complete all three tasks. Incidentally, teachers had not managed to fully complete or discuss individual assignments due to time issues, but overall, teachers had not reported major deviations.

### 4.2 Understanding score, blind score & no concept score

Table 3 presents the comparisons of all the models that were estimated.

From Table 3, it can be inferred that for the Understanding Scores and the No Concept Scores there is an effect of the intervention (for both Model 2 outperforms Model 1, which is reflected in the higher elpd values for Model 2 as compared to Model 1), although there is no effect for the Blind Scores (more complex models do not outperform Model 1, reflected in Model 1 having the highest elpd value). Table 4 gives the parameter estimates and the 95% highest density intervals (HDI) for the core variables. For the variable Blind Score, no parameter estimates related to the effect of Group 1 are included. For the two other variables the best fitting models include a parameter for the effect of the dummy variable Group 1 at measurement occasion 2.

With the switching replications design the effect of the intervention is replicated by design. For Group 1 students who received the intervention between measurement occasion 1 and measurement occasion 2 the effect of the intervention is reflected in the interaction term between measurement occasion 2 and the Group 1 dummy variable (Occ2 * Group1). The

**Table 3. Model comparison for the three core variables: Understanding score, blind score and no concept score.**

| *Understanding score* | | | *Blind score* | | | *No Concept score* | | |
|---|---|---|---|---|---|---|---|---|
| **Model** | $\Delta\widehat{elpd}$ | $\widehat{elpd}$ | **Model** | $\Delta\widehat{elpd}$ | $\widehat{elpd}$ | **Model** | $\Delta\widehat{elpd}$ | $\widehat{elpd}$ |
| *Step 1* | | | *Step 1* | | | *Step 1* | | |
| Model 2 | - | -1202.82 | Model 1 | - | -682.69 | Model 2 | - | -962.83 |
| Model 3 | -0.90 (0.54) | -1203.72 | Model 2 | -0.69 (0.88) | -683.38 | Model 3 | -1.11 (0.23) | -963.93 |
| Model 4 | -1.21 (0.40) | -1204.03 | Model 3 | -1.41 (1.04) | -684.10 | Model 4 | -1.40 (0.39) | -964.23 |
| Model 1 | -8.34 (4.52) | -1211.16 | Model 4 | -1.42 (1.19) | -684.12 | Model 1 | -7.31 (3.76) | -970.14 |
| *Step 2* | | | *Step 2* | | | *Step 2* | | |
| Model 5 | - | -1201.81 | Model 1 | - | -682.69 | Model 6 | - | -961.86 |
| Model 6 | -0.15 (0.29) | -1201.96 | Model 5 | -1.49 (1.00) | -684.18 | Model 2 | -0.97 (2.15) | -962.83 |
| Model 2 | -1.01 (2.19) | -1202.82 | Model 6 | -3.20 (2.08) | -685.89 | Model 5 | -1.95 (1.64) | -963.80 |

**Table 4. Mean and 95% Highest Density Interval (HDI) for the posterior distributions of the parameter estimates based on the best fitting models for understanding score (Model 5), blind score (Model 1) and no concept score (Model 6).**

|  | Understanding Score [95% HDI] | Blind Score [95% HDI] | No concept Score [95% HDI] |
|---|---|---|---|
| Intercept Occ1 | 10.149 [9.723, 10.586] | 1.1397 [1.242, 1.558] | 4.128 [3.865, 4.382] |
| Intercept Occ2 | 11.132 [10.649, 11.584] | 1.525 [1.366, 1.675] | 3.553 [3.255, 3.870] |
| Intercept Occ3 | 13.197 [12.681, 13.706] | 1.296 [1.169, 1.427] | 2.399 [2.133, 2.657] |
| Occ2 * Group1 | 2.247 [1.417, 3.061] | -,- | -1.427 [-1.966, -0.895] |
| Gender (0 = boy; 1 = girl) | 0.932 [0.197, 1.642] | -,- | -0.542 [-0.952, -0.113] |
| Version 2 | -,- | -,- | 1.800 [1.209, 2.378] |
| Version 3 | -,- | -,- | 0.362 [-0.245, 0.935] |
| Δ Intercept Occ3 & Intercept Occ2 | 2.064 [1.373, 2.730] |  | -1.152[-1.539, -0.759] |

note: -,- parameter not included in the best model; Occ = Measurement occasion.

95% of most credible estimates for this parameter concerning the understanding score are situated between 1.42 and 3.06 indicating that students in this group score higher than students in Group 2 who did not receive the intervention at that measurement occasion. The effect of the intervention for Group 2 students can be derived from the difference between measurement occasion 3 and measurement 3 occasion (Δ Intercept Occ3 & Intercept Occ2), indicating that 95% most credible estimates of the intervention for Group 2 students are situated between 1.37 and 2.73. In Fig 2 the posterior distribution for the estimated means for both groups on each occasion are plotted. From this plot (see panel A) it can be read that all post-intervention means for the understanding score are higher than the pre-intervention scores. Based on the model we also calculated the posterior distribution of the Cohen's *d* (see Fig 3) for the effect of the intervention in each group. For the understanding score (see panel A of Fig 3) we learn that the effect size of the intervention in both groups is most probably situated between medium and large (with a higher probability for large effect sizes for Group 1).

For the No concept score we learn that the intervention has resulted in lower scores. The 95% most credible estimates of effect of the intervention for Group 1 are situated between -1.97 and -0.90 and for Group 2 between -1.54 and -0.76. In Fig 2 this is reflected in the systematic lower estimated means for measurements after the intervention. The posterior distribution of the effect sizes for No concept scores (see panel B of Fig 3) indicate that the most probable effect sizes are situated between a medium and large effect size (with a higher probability for large effect sizes for Group 1).

### 4.4 Qualitative analysis of intervention assignments

**4.4.1. Key assignment 1: Explain in your own words.** In key assignment 1, students had to explain why some sentences contain objects, whereas others do not (cf. Appendix C in S1 File). This task was designed to see whether students could adequately describe what they had just learned about valency in their own words. In Table 5, we will summarize the relevant axial codes, illustrate them with typical examples from the data and then discuss them. Apart from codes related to missing or unusable data, the analysis revolved around the structure of responses, and the language used to refer to important concepts (*grammatical metalanguage* vs. *everyday language*). Finally, some codes related to the students' level of understanding. For this task, students wrote on average 11.75 words per response (*SD* = 6.91).

What stands out from Table 5, is that a majority of the 146 returned tasks revealed some level of understanding (87.7%), either at the level of *partial understanding* (54.8%) or *full understanding* (32.9%). A minority of students did not manage to formulate an adequate

**Table 5. Axial codes and subcodes related to key assigment 1 (N = 196 students; N = 146 performed and readable tasks).**

| Axial code | Subcodes | Definition of subcode | Typical example from data (translated) | N |
|---|---|---|---|---|
| **Unusable** | **Unreadable** | **Response that could not be read** | **NA** | **3** |
| | **Left blank** | **Answers left blank** | **NA** | **47** |
| TOTAL | | | | 50 |
| Structure of response | Single statement | Single statements without further elaboration | It has to do with the role of the verb. | 101 |
| | Single statement + elaboration | Single statement which is elaborated upon, explaining the previous statement | Sometimes verbs cannot function if it is not being told with whom or what you do something, because the verb is then incomplete. | 25 |
| | Single statement + elaboration + example | Single statement which is elaborated upon, illustrated with an example | Some verbs are hard to combine with direct and indirect objects (to hockey, to rain). To give on the other hand is an example of a verb with many possibilities. Some things you can do to or for someone, and others, you can't. It has to do with valency. | 4 |
| | Single statement + example | Single statement illustrated with an example | You cannot place something (direct/indirect object) with every verb, for example with the verb *turnen* ('do gymnastics'). You can't turn something or someone. | 4 |
| | Example | Reply containing just an example | You can't 'swim something', but you can 'take something away from someone'. | 2 |
| | Other | Response that cannot be characterized in terms of structure easily, e.g., one or two word responses | Not everything. | 10 |
| TOTAL | | | | 146 |
| Grammatical metalanguage | Direct object | Explicit mentions of direct object | Some verbs cannot have *direct objects*. | 40 |
| | Indirect object | Explicit mentions of indirect object | Some verbs cannot have *indirect objects*. | 40 |
| | Verb | Explicit mentions of verb | Some *verbs* cannot have objects. | 115 |
| | Valency | Explicit mentions of valency | Some verbs have greater *valency*, allowing for more possibilities. | 11 |
| | Role (of the verb) | Explicit mentions of role(s) (of the verb) | Some verbs can have *multiple roles*. | 22 |
| | Preposition | Explicit mentions of preposition | Some verbs need the preposition *aan*, creating an indirect object. | 5 |
| | Subject | Explicit mentions of subject | Some verbs can only have a *subject*, some can also have a direct object and some can have indirect objects. | 3 |
| | Other | Explicit mentions of any other metalanguage | NA | 27 |
| TOTAL | | | | 263 |
| Concepts in everyday language | Role | Using the word 'role' in the generic sense instead of the linguistic sense | It has to do with *the role of the verb*. (i.e., *role* is used to replace *function* here). | 11 |
| | Possibilities of the verb | Expression that verbs have possibilities | You can do more with one verb then with another. | 29 |
| | Direct object (implicit) | Implicit reference to direct objects | Someone *who undergoes something*. | 9 |
| | Indirect object (implicit) | Implicit reference to indirect objects | With the verb 'to take away', you won't just get a question about the what, but also about the '*from whom*'. | 15 |
| Understanding | No understanding | Response makes no sense, or is conceptually flawed | You won't always need them. / I don't understand any of it. | 14 |
| | Partial understanding | Some form of partial understanding is conveyed | It has to do with the role of the verb. | 80 |
| | Full understanding | Response makes clear that the student has understood the core idea, using appropriate metalanguage | Because there can be a certain number of roles for each verb. Not every verb has the same number (valency of a verb). | 48 |
| | Unassessable | Level of understanding cannot be determined adequately | NA | 4 |
| TOTAL | | | | 146 |
| TOTAL *N* of codes | | | | 604 |

response to the task (9.6%). It is important to note, however, that within the category of *full understanding*, there are some quality differences between the responses, although we will not elaborate on them here.

There appears to be a relationship between the structure of students' responses and their level of understanding. For example, most of the students who showed signs of *partial understanding* responded using single statements (62/80, 77.5%). From the 14 students showing no signs of understanding, 100% used either single statements or provided only an example–one of them overtly expressing that he had not understood. From the students showing *full understanding*, 32 (66.7%) turned to single statements. This is still a two thirds majority, but the overall percentage is lower.

In terms of their (meta)concept use, students frequently included direct and indirect objects in their response, although this does not come as a surprise, as these concepts were prompted in the question. Many students (78.8%) do use the concept of *verb* in their response (unprompted), indicating that students have seen the importance of verbs for the occurrence of objects, which lies at the heart of understanding valency. 22.6 percent of all students included an explicit reference to the (new) concept of valency, or to 'roles of the verb'. The concept of *role* is also used in a less metalinguistic way by some, meaning that they will use the everyday term (in the singular) rather than the metalinguistic term (usually in the plural).

**4.4.2 Key assignment 2: Multimodal task.** In key assignment 2, students were asked to respond to a humanitarian aid poster showing reduced valency (cf. Appendix C in S1 File). Students were asked to indicate what stood out to them on the poster in terms of valency (explicitly prompted), and to think about the rhetorical effect of this valency use. Table 6 illustrates the main axial codes. These codes predominantly focus on identifying the issue of reduced valency, and the rhetorical effect this yields. On average, students wrote 20.9 words for this task (*SD* = 16.44). From Table 6, it can be inferred that 45.1% of all responses failed to detect an issue related to valency in the humanitarian aid poster. Some students (15.5%) had identified a valency related issue, but did so not completely adequately, making errors in their description of the concept. 47 students (33.1%) had managed to recognize the valency issue,

**Table 6. Axial codes and subcodes related to key assignment 2 (N = 196, N = 142 performed and readable tasks).**

| Axial code | Subcodes | Definition of subcode | Typical example from data | N |
|---|---|---|---|---|
| **Unusable** | **Makes no sense** | **Answer that cannot be interpreted properly** | **Then you have no connections. It is very direct and therefore very big.** | **11** |
| | Left blank | Answers left blank | NA | 43 |
| TOTAL | | | | 54 |
| Dealing with valency | Valency issue recognized but inadequately | Response deals with valency issue (implicitly or explicitly) but does so inadequately (e.g., by making errors in the assessment of the valency issue). | They don't provide any roles apart from the main verb. | 22 |
| | Valency issue recognized but not linked to related appropriate rhetorical effect | Response adequately identifies the valency issue (implicitly or explicitly), but fails to make a connection with the rhetorical consequence | You are missing the roles with some of the verbs. It is more of an order, and it is clearer. | 9 |
| | Valency issue recognized and related to related rhetorical effect | Response adequately identifying the valency issue and adequately relating it to an appropriate rhetorical effect | All of the text is in the imperative. And it is very unspecific what it is that you need to give. You would probably realize that it is money you are supposed to give. I find this very useful. It will make you add things on your own, and if you really want to know more about it, you will look it up. Because of the context of the poster you can infer for yourself that it is about money. They have thus left out many roles. | 47 |
| | Rhetorical effect only or rhetorical effect based on other grammatical feature | Response only deals with a rhetorical effect or links a different grammatical issue to a rhetorical effect | The adverbial is missing. It sounds more urgent that way. | 64 |
| TOTAL | | | | 142 |
| TOTAL *N* of codes | | | | 196 |

**Table 7. Axial codes and subcodes related to key assignment 3 (N = 196, T1: N = 123; T2: N = 121 performed and readable tasks).**

| Axial code | Subcodes | Definition of subcode | Example (T1, T2) | N T1 | N T2 |
|---|---|---|---|---|---|
| Unusable | Unreadable | Answers which cannot be read | NA | 1 | 0 |
| | Left blank | Answers left blank | NA | 72 | 75 |
| TOTAL | | | | 73 | 75 |
| Exclusion | Exclusion without argumentation | Response in which an odd one-out is chosen without any argumentation. | T1: 'Roken' is the odd one-out. | 6 | 6 |
| | | | T2: 'op een tractor' is the odd one-out. | | |
| | Exclusion based on appropriate grammatical argument | Response in which odd one-out is chosen based on a grammatical argument that is true and appropriate. | T1: 'Groeien' is the odd one-out, because you can add a direct object to the other two verbs. | 78 | 39 |
| | | | T2: 'Op een tractor' is the odd one-out, because the other two are essential parts of speech of the verbs that are in the sentences. | | |
| | Exclusion based on inappropriate grammatical argument | Response in which odd one-out is chosen based on a false or untrue grammatical argument. | T1: 'Krijgen' is the odd one-out, because that needs a preposition. | 33 | 41 |
| | | | T2: 'Op een tractor' is the odd one-out, because the other two are objects and 'op een tractor' is an adverbial. | | |
| | Exclusion based on non-grammatical argument | Response in which odd one-out is chosen based on a non-grammatical argument. | T1: 'roken' is the odd one-out, because smoking is not healthy. | 4 | 35 |
| | | | T2: 'Op een tractor' is the odd one-out, because the other two are persons, but not the tractor. | | |
| TOTAL | | | | 121 | 121 |
| Response | Formulation based on prescribed format | Response based on the prescribed format from the example: X is the odd one-out, because the other two . . . | T2: 'Op een tractor' is the odd one-out, *because the other two* are bound to the verb. | 86 | 91 |
| Grammatical metalanguage* | Adverbial | Explicit mention of adverbial | T1: NA | 0 | 10 |
| | | | T2: (. . .) because the other two are *adverbials*. | | |
| | Agency (implicit) | Implicit reference to agency | T1: 'Groeien' and 'Krijgen' *you do yourself*, whereas 'krijgen' *is being done by someone else*. | 19 | 0 |
| | | | T2: NA | | |
| | Direct object | Explicit reference to direct object | T1: 'Groeien' is the odd one-out, because you can add a *direct object* to the other two verbs. | 26 | 21 |
| | | | T2: 'Grandma' is the odd one-out, because the other two are *direct objects*. | | |
| | Optionality | Mentions or indicates that a part of speech is optional or mandatory | T1: NA | 0 | 28 |
| | | | T2: (. . .) because the other two *cannot be left out*. | | |
| | Preposition | Explicit reference to preposition | T1: 'Krijgen' is the odd one-out, because that needs a *preposition*. | 9 | 6 |
| | | | T2: (. . .) because the other two don't have fixed *prepositions*. | | |
| | Role / valency | Explicit reference to role or valency | T1: 'Groeien' is the odd one-out, because the other two can have *two roles*, and 'groeien' cannot. | 13 | 1 |
| | | | T2: (. . .) because the other two *have fixed roles*. | | |
| | Role (implicit) | Implicit reference to grammatical role(s) | T1: (. . .) because with the other two, you need to know *what you receive or what you smoke*. | 11 | 0 |
| | | | T2: NA | | |
| | Valency pattern testing | Tests the valency pattern of a verb | T1: 'Groeien' is the odd one-out, because with the other two, you can say: *'Ik krijg iets'/ 'Ik rook iets'*. | 19 | 1 |
| | | | T2: 'Someone sees something', 'someone bakes something', but not: 'someone drives something'. | | |

(*Continued*)

**Table 7.** (Continued)

| Axial code | Subcodes | Definition of subcode | Example (T1, T2) | N T1 | N T2 |
|---|---|---|---|---|---|
| | Verb | Explicit reference to verb | T1: 'Groeien' is the odd one-out, because you can add a direct object to the other two *verbs*. | 2 | 7 |
| | | | T2: (. . .) because the other two are bound by the *verb*. | | |
| | Weak verb | Explicit reference to weak verb | T1: 'Krijgen' is the odd one-out, because the other two are weak verbs. | 7 | 0 |
| | | | T2: NA | | |
| | Other | Explicit reference to other grammatical concepts | NA | 14 | 28 |
| TOTAL | | | | 120 | 92 |
| TOTAL *N* of codes | | | | 400 | 379 |

* Note Only codes mentioned >5 times have been taken into account.

and relate it to some (more or less) appropriate rhetorical effect. For example, the following response relates the reduced valency to an appropriate rhetorical effect: *'It says: give now!', but it does not say what needs to be given. This makes people think not just about money, but more about how bad it is out there, making it easier for people to make a donation'*. By comparison, the following response has more specifically indicated valency reduction using grammatical terms, but links this to a less obvious rhetorical effect: *'Very few roles are being used in this poster, only one or two. This makes people feel more forced to actually help and it reads easier.'*

**4.4.3 Key assignment 3: Two odd one-out tasks.** In key assignment 3, students tackled two odd one-out tasks, Task 1 and Task 2 (Appendix C in S1 File). The first odd one-out task focused on verbs directly, making it more likely that students would use valency or think based

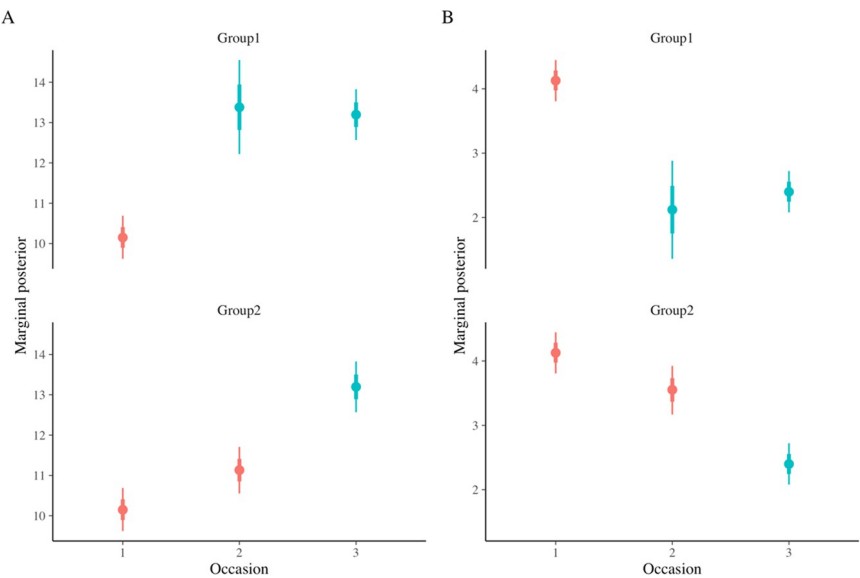

**Fig 2.** Median, 66% and 95% CI for the posterior distribution of means for both groups (Group 1 = intervention between Occ1 & Occ2; Group 2 = intervention between Occ2 & Occ3) on Understanding score (A) and No Concept score (B). Blue intervals indicate means after implementation of the intervention; red intervals indicate means before the implementation of the intervention.

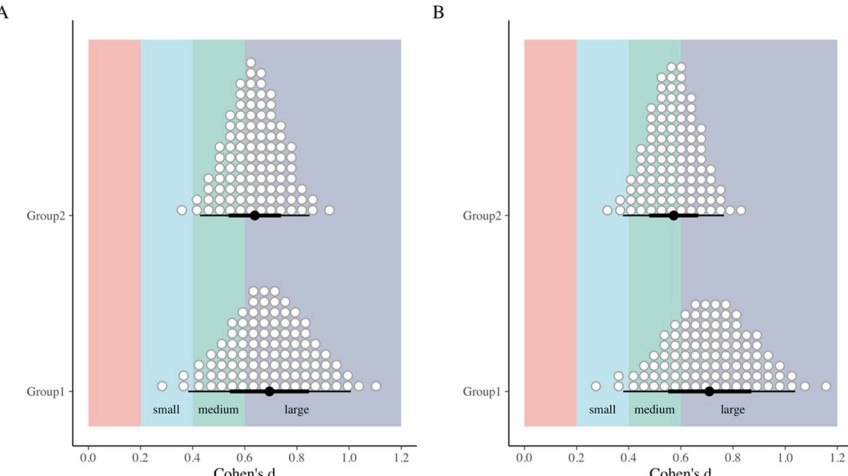

**Fig 3.** Posterior distribution for the absolute effect sizes (Cohen's d) of the intervention for Understanding scores (A) and No Concept scores (B) with the median and 66% and 95% CI (Effect size labels according to Calin-Jageman and Cumming [85]). Group 1 = intervention between Occ1 & Occ2; Group 2 = intervention between Occ2 & Occ3.

on the concept of valency to determine which verb is the odd one-out (e.g., X is the odd one-out, because the other two can have two roles / can have direct objects). The second task could also be tackled based on valency, but more indirectly. Both tasks allowed for different grammatical reasons of exclusions as well (e.g., X is the odd one-out, because the other two are weak verbs).

In Task 1, students wrote 15.78 words on average (*SD* = 4.87); in Task 2, students wrote 14.54 words on average (*SD* = 4.31). In the coding (see Table 7), we particularly noted students' type of argumentation (*Exclusion*). In addition, we verified how many students had followed the prescribed format (*Response*) and what grammatical terms they used in their reasoning (*Grammatical metalanguage*). From Table 7, it can be inferred that Task 2 was grammatically more complicated for students, since there was a substantial difference between the two tasks regarding the number of times that students had provided an appropriate grammatical argument to exclude an option (64.5% for Task 1, versus 32.2% for Task 2). Upon closer inspection, this difference might be partly caused by terminological confusion between *role* and *object*, as 12 students (9.9%) had excluded one option based on the premise that the other two were *objects*, which is inaccurate, but it is not unlikely they may have confused the concept of object with that of role. 9 additional students had done the same for *direct objects* specifically; it is, therefore, possible that 21 students (17.4%) have in fact given a conceptually sound reason for exclusion, using the wrong term. The number of exclusions based on non-grammatical arguments was, however, much higher for Task 2 (28.9%) than for Task 1 (3.3%). In their responses for Task 1, students showed signs of understanding valency or roles 57% of the time (either by directly using these concepts, or by elaborating on the presence of objects); for Task 2, this percentage was much lower (24.8%).

## 5. Discussion

### 5.1 Interpretation of main findings

The current study set out to establish whether a metaconceptual intervention could positively impact students' L1 grammatical understanding. In addition, it investigated whether potential progress could be attributed to a testing effect. As its secondary objectives, the study looked

into (a) whether blind concept use was a negative byproduct of the intervention, (b) whether the intervention would influence students to favor explanations for grammatical problems which use relevant grammatical terminology over everyday terminology, and (c) what key assignments from the intervention could reveal about students' grammatical understanding.

Our main research question could be answered positively: metaconceptual interventions can have a strong impact on students' level of grammatical understanding. Following the interpretation of effect sizes by Calin-Jageman and Cumming [85], we conclude from the Bayesian model estimation that the intervention overall has at least a medium effect but most probably a large effect (>0.6). This effect becomes particularly meaningful in light of the shortness of the intervention, which was only four lessons. In addition, since the control group did not progress between measurement occasions 1 and 2, it can be inferred that students' progress cannot be attributed to a testing effect. The current study thus provides more robust evidence in favor of metaconceptual approaches to grammar learning, following up on previous work [55, 56] in which it was shown that metaconceptual approaches might be beneficial for grammatical reasoning ability.

While Van Rijt et al. [56] found that students tended to use some (meta)concepts in their reasoning blindly as a negative byproduct of the intervention, the current study finds no evidence for this. Students did sometimes opt for alternatives in which a blind concept was presented, but the intervention had no influence on their preference for blind or non-blind concept use. This might be due to the fact that in Van Rijt et al. [56] students were prompted to write reasonings (i.e., using their own words), which might evoke more blind concept use than in the current study, in which students did not actively produce lengthy reasonings, but only had to choose the best alternative. An alternative explanation might be that, as stated in the introduction, the current intervention only targeted one metaconcept (valency), while the Van Rijt et al. [56] intervention targeted four metaconcepts. It would seem likely that the chances of blind (meta)concept use increase when there are more metaconcepts that have to be learned in the first place. Differences in conceptual content covered in the interventions could thus account for differences in blind concept use between the current study and Van Rijt et al.'s [56].

The intervention had a positive effect on students' preference for explanations for grammatical problems which use relevant grammatical terminology over everyday terminology, with effect sizes that are situated between medium and large. The result is interesting in light of previous work [56] in which it was indicated that teachers favor explicit grammatical (meta)concept use in students' grammatical reasoning rather than everyday descriptions of grammatical concepts (i.e., implicit or everyday concept use). The intervention thus appears to stimulate students to think about grammatical (meta)concepts and can subsequently improve on their grammatical thinking.

The qualitative analyses of intervention assignments shed a potentially different light on students' conceptual learning. Since understanding comes in degrees, it is likely that several students are in the process of acquiring the new terms and grasping the concepts that underlie them. This means that they may still need to figure out how to use such terms adequately, and this may lead to cases of blind concept use, or formulations favoring everyday concepts over metalinguistic concepts. For example, some students used the concept of *role* in an everyday sense rather than in a metalinguistic sense, indicating that they have not yet fully grasped the grammatical concept yet. Their use of the word *role* may therefore be seen as an indication of a limited understanding, although, encouragingly, it may also mean that students are on a conceptual journey, a part of which is to struggle with new terminology and concepts. This struggle between everyday concepts and subject-specific concepts has also been observed in science education, where students have, for example, been found to struggle with learning about the

scientific concept of 'energy' [89], which also has an everyday equivalent. A similar observation has been made in students' use of historical concepts in history education. Not only do students typically use historical concepts in a very limited way when reasoning about history, but according to Havekes (p. 69), they also 'apply everyday discourse to describe a historical concept, such as 'changes in the church' instead of 'reformation'' [66]. These examples suggest that an integral part of conceptual learning might be that students have to learn how to deal with tensions between everyday concepts on the one hand and scientific concepts that they are expected to master in the context of a school subject on the other (see also the distinction between *Basic Interpersonal Communicative Skills (BICS)* and *Cognitive Academic Language Proficiency (CALP)* [90]). In this sense, the current study shows that there might be parallels between conceptual learning in linguistics education and conceptual learning in other fields. We leave further exploration of this matter open for future research.

Another case in point would be that students experience some conceptual confusion as a result of the traditional ways in which grammar is usually taught. Students might not be accustomed to reason within the context of grammar instruction, which is usually mostly concerned with superficial parsing exercises [34, 35]. This pertains to the meaning of the very concept of *grammar* in this study, as the everyday use of this concept is rather different from the linguistic concept of grammar (with the former being mostly related to prescriptivism and language norms, and the latter being mostly associated with descriptivism and cognitive capacity). Students have been shown to hold views on grammar that are more related to the everyday meaning than to the linguistic meaning of the word [91, 92].

When interpreting the results of this study, it can be observed that the effect size of the intervention appears to be larger overall in group 1. As this difference can possibly be accounted for by (quality) differences between teachers, future studies would do well to incorporate this variable into the study's design.

The qualitative data show that some of the key assignments need more attention, both from the teachers guiding such assignments in practice and from those designing such tasks. For example, in key assignment 1, the majority of students responded in only one short statement. And while some of the students manage to do this in a reasonable way, more elaborated responses will provide more opportunities for teachers to verify whether grammatical understanding has taken place, and if so, to what degree. Designers of such tasks should therefore put more scaffolds in the assignment (e.g., word minimums) or provide examples of adequate response types. Teachers in their turn can capitalize on the effect of the intervention more if they can find ways to let their students answer the questions more elaborately, and if they can learn how to scaffold students' responses more adequately.

## 5.2 Limitations

The current study is not without limitations. First of all, there is no data to shed more light on students' reasoning (for example, from think aloud protocols), that could deepen our understanding of students' understanding both within and outside of the intervention. Additionally, while we deliberately refrained from classroom observations, greater detail on how the teachers had performed would have been a welcome addition to this study. Moreover, the study has provided interesting insights related to the workings of a metaconceptual intervention, but such an approach cannot be directly compared to traditional grammar teaching practices, as we deliberately did not contrast these two treatments. And finally, while the current study has greatly increased the number of participants compared to previous intervention studies, its findings cannot yet be fully generalized to other educational settings. Future research would do well to include a larger qualitative component, e.g., letting students think aloud in post-hoc

interviews or think aloud protocols. Such data is only useful if think aloud protocols are applied in a way that avoids the pitfall of cherry picking. One way to achieve this is to randomize which students participate in the protocol; another is to pre-structure the protocol, focusing only on a few pre-determined variables, so that all of the data can be coded, rather than a few pieces of data that might match the researchers' needs. A final recommendation regarding the role of thinking aloud protocols is to use them retrospectively (rather than simultaneously to the task), as this increases their validity [93].

In addition, the role of the teacher needs further exploration. It would, for example, be important to investigate to what extent the teachers' own grammatical understanding impacts on the students' grammatical learning, and how their epistemic beliefs towards grammar influence the way in which they handle metaconceptual interventions [60, 94]. Finally, follow-up research might compare traditional parsing treatments to metaconceptual treatments for a clearer idea of the added benefit of metaconcepts compared to traditional teaching.

## 6. Conclusion

The current study has demonstrated that metaconceptual grammar teaching has positive effects on students' grammatical understanding, while finding no evidence for increased 'blind' concept use as a negative by-product of such grammar teaching. Cases of blind concept use that do occur might not be completely negative at all. Rather, cases of blind concept use might reflect a necessary stage in some students' conceptual understanding. In line with this development, the study found that a metaconceptual approach has a positive influence on students' explicit concept use when explaining grammatical problems.

While there is still a great need for additional research, the current study provides more robust evidence in favor of interventions that capitalize on underlying linguistic metaconcepts to foster grammatical understanding.

## Supporting information

**S1 File.**
(DOCX)

## Author Contributions

**Conceptualization:** Jimmy van Rijt, Peter-Arno Coppen.

**Data curation:** Jimmy van Rijt.

**Formal analysis:** Jimmy van Rijt, Debra Myhill, Sven De Maeyer, Peter-Arno Coppen.

**Funding acquisition:** Jimmy van Rijt, Peter-Arno Coppen.

**Investigation:** Jimmy van Rijt, Debra Myhill, Peter-Arno Coppen.

**Methodology:** Jimmy van Rijt, Debra Myhill, Sven De Maeyer, Peter-Arno Coppen.

**Project administration:** Jimmy van Rijt, Peter-Arno Coppen.

**Resources:** Jimmy van Rijt.

**Software:** Jimmy van Rijt, Sven De Maeyer.

**Supervision:** Peter-Arno Coppen.

**Validation:** Jimmy van Rijt.

**Visualization:** Jimmy van Rijt, Sven De Maeyer.

**Writing – original draft:** Jimmy van Rijt.

**Writing – review & editing:** Jimmy van Rijt, Debra Myhill, Sven De Maeyer, Peter-Arno Coppen.

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
