## [Decision Letter · Decision Letter 0]

11 Nov 2021

PONE-D-21-18569Linguistic metaconcepts can improve grammatical understanding in L1 education: evidence from a Dutch quasi-experimental studyPLOS ONE

Dear Dr. Dr. Jimmy van Rijt

Thank you for submitting your manuscript to PLOS ONE. After careful consideration, we feel that it has merit but does not fully meet PLOS ONE’s publication criteria as it currently stands. Therefore, we invite you to submit a revised version of the manuscript that addresses the points raised during the review process.

Both reviewers were happy with your paper and the research reported. In my view, there is no major issue that would prevent publication. However, please, make sure you address *all* comments and concerns raised by the reviewers at the best of your capacity. They will certainly help you improve your manuscript and strengthen its impact. 

We look forward to receiving your revised manuscript.

Kind regards,

Christina Manouilidou, Ph.D

Academic Editor

PLOS ONE

Journal Requirements:

2. Peer review at PLOS ONE is not double-blinded (https://journals.plos.org/plosone/s/editorial-and-peer-review-process). For this reason, authors should include in the revised manuscript all the information removed for blind review, including names of universities.

Reviewers' comments:

Reviewer's Responses to Questions

**Comments to the Author**

1. Is the manuscript technically sound, and do the data support the conclusions?

Reviewer #1: Yes

Reviewer #2: Yes

2. Has the statistical analysis been performed appropriately and rigorously? 

Reviewer #1: Yes

Reviewer #2: Yes

3. Have the authors made all data underlying the findings in their manuscript fully available?

Reviewer #1: Yes

Reviewer #2: Yes

4. Is the manuscript presented in an intelligible fashion and written in standard English?

Reviewer #1: Yes

Reviewer #2: Yes

5. Review Comments to the Author

Reviewer #1: This is a very interesting and relevant study. The research problem is very well-motivated, the objectives are stated clearly, the methodological design seems adequate and robust, the number of participants recruited for the experiment is also impressive. Furthermore, the results provide encouraging evidence concerning the positive impact of meta conceptual grammar teaching on grammatical understanding.

In general, I feel like the paper has merit and should be published once some adjustments/corrections are made and some points are more appropriately addressed. The paper is very well-written; however, some textual issues deserve attention. I would recommend a major revision of punctuation, for instance.

General comments are presented in the following:

###Introduction###

Page 5: I feel like these two parts within the paragraphs should be better articulated, perhaps changing their order?:

"This type of grammar teaching, focusing more on the mechanic articulation [...]" and "On the other hand, contextualized forms of grammar teaching [...]".

Page 12: You should not assume all your readers are aware of what "valency" is. Please, provide an explicit definition and, if possible, some examples.

Page 13: "Teachers were told that they could

deviate from the specifics of the intervention to a limited degree if the situation called for it [...]": Okay, did they deviate much? I suggest you anticipate something about it or just say that it will be described further.

Page 14: "And finally, one of the answers used no grammatical terminology whatsoever, but instead represented an answer

based on everyday language and experiences.": How much would students score in this case? say it (like you did before), even if it is in the table.

Page 16: "Additional Bonferroni post hoc tests": Additional Bonferroni adjusted post hoc tests

Pages 18-20: Quantitative analysis: I suggest you try to link the tests you have performed with the research questions you are trying to answer. It may be easier to follow, especially for non-experts.

Page 18: "At measurement moment 3": 3 or 2?

###Results###

Page 24: "From Table 4, it can be inferred that [...]": could you please guide your reader in the reading of the table? What specifically supports your claims? is it an increase/reduction in the scores? Say it. You do it properly in other parts.

Page 24: Table 5: Are these values in the natural log?

It would be very interesting to present some proposed verbal equivalents for likelihood ratios, like in a table. It would help the reading and the understanding of your results considerably. Not everyone is familiarized with the Bayesian analysis and the way the outcomes are reported. It would be much easier to grasp Table 5, for instance.

Page 26: Figure 3: Although it may seem repetitive, you need to remind your readers what group 1 and group 2 are. It is easy to forget.

Table 8 and 9: You need to separate your rows better. It is very confusing the way it is. We need to make an effort to understand when things start or end in the "Definition of subcode" column.

The title of Table 9 is out of the table.

###Discussion###

Page 38: Although the researchers have a good sense of the study limitations, it is odd that you start the discussion by first stating it. Why don't you start by discussing your main findings instead? I suggest you leave the study limitations for last, before the conclusions*.

In the introduction (page 12, second paragraph), you report that one of the main differences between your study and the one conducted by [Van Rijt et al.] was that rather than focusing on four different linguistic

metaconcepts, the intervention focused on just one: valency. I thought this difference should be highlighted and better exploited in the discussion, especially considering that you did not find a negative side-effect in your study, but they did. One would expect that having a larger number of linguistic concepts/terminologies to handle would impact the students' performance in general. You refer to it in your introduction when saying that "students would have the opportunity to process the newly taught information more effectively, limiting the

amount of new linguistic terminology", however, you do not explore this in your discussion, you only suggest (last paragraph of page 39) that this may be because in [Van Rijt et al.] students were prompted to writing reasonings.

Page 41: *No conclusions section? I find it interesting to provide your reader a summary of your primary outcomes.

###Some textual issues###

Page 3:

‘a central good -> a central good

And while there appears to be a growing -> Furthermore, while there appears...

What is becoming more and more accepted, is the idea that -> remove the comma.

Page 4:

Strenghtening -> Strengthening

the question how such -> of how

And while many teachers consider grammar teaching -> Moreover, while many teachers...

text books -> textbooks

Page 5:

therefore likely to impact on both areas -> to impact both areas

Page 6:

text books -> textbooks

And while the metaconcept of valency is -> In addition, while...

as ‘higher order concepts -> as higher-order concepts

higher order -> higher-order

Page 7:

Describing concepts in their own words in their grammatical reasoning (implicit concept use), was not positively -> Remove the comma

pre- and posttest -> pre-and post-test

judgement -> judgment

Page 8:

judgements -> judgments

Such odd one out -> Such an odd one out

Page 9: Add a period after "as was demonstrated by Van Rijt, De Swart, Wijnands and Coppen [55]"

Page 10:

potential efficiacy -> efficacy

Page 11:

they were not allowed to teach any grammar, to ensure that -> no comma needed here

Page 12:

in some degree -> to some degree

For the intervention a student assignment booklet -> For the intervention, a student assignment (comma added)

they were given an elaborate instruction manual, containing a background -> remove comma

multiple choice -> multiple-choice

Page 13: Add a period after "following the recommendation of Rijlaarsdam et al. [72]"

Page 15:

multiple choice options -> multiple-choice options

From this we concluded that -> From this, we concluded that

Page 17:

had been answered to -> had been answered

Across the three variables and measurements no extreme outliers were found -> Across the three variables and measurements, no extreme outliers were found (comma added)

At M3, the attrition rate was higher, due -> no comma needed

modelling -> modeling

Page 18:

(M1i, M2i and M3i) -> (M1i, M2i, and M3i)

variance covariance -> variance-covariance

In Model 2 we add -> In Model 2, we add...

Group 1 will score different -> differently

Page 19:

In a final step we elaborated -> In a final step, we elaborated

For the inferences we visualized the posterior probability -> For the inferences, we visualized...

large effects respectively, -> large effects, respectively,

0.2, 0.4 and 0.6 -> 0.2, 0.4, and 0.6

analyzed in more detail, to investigate -> remove comma

both individually and, later, together. -> both individually and later together.

Page 24:

For the two other variables the best fitting -> For the two other variables, the best fitting

Page 25:

indicating that 95% most credible -> indicating that 95% of most credible

The posterior distribution of the effect sizes indicate -> The posterior distribution of the effect sizes indicates

Page 25: Add a period after "[...] means before the

implementation of the intervention" in Figure 2's title.

Page 33:

one out tasks -> one-out tasks

regarding the amount of times -> regarding the number of times

it is therefore possible that -> it is, therefore, possible that

it is therefore possible that 21 students (17.4%) have in fact given -> it is, therefore, possible that 21 students (17.4%) have, in fact, given

The number of exclusions based on non-grammatical arguments was however much higher -> The number of exclusions based on non-grammatical arguments was, however, much higher

Page 38:

(for example from think aloud protocols) -> (for example, from think-aloud protocols)

It would for example be important -> It would, for example, be important

positively impact on students’ L1 -> positively impact students’ L1

Page 39:

bayesian model -> Bayesian model

has a at least -> has at least

which capitalize -> that capitalize

Reviewer #2: I'm already familiar with the research coming from this group, and I think it's one of the best research programs in pedagogical linguistics at the moment. The present paper presents original and new findings, based on previous studies. Importantly, the main finding (at least for me) is that students' blind concept use is actually an integral part of a gradual understanding of grammatical concepts. This is extremely interesting and also relevant for other domains of teaching and learning. Given my positive evaluation of this manuscript (and of the underlying research program as a whole), I only have three very minor comments for the authors:

1) Students start out by using strategies like 'name dropping' etc. before they are able to use grammatical concepts appropriately. Is there a general sequential model for this in the educational sciences/in learning theory? The 'fake-it-till-you-make-it' strategy seems to be a common developmental step in other domains too, so I'm wondering whether the authors could clarify this aspect of their findings even further, making their research even more relevant for a journal like PLOS ONE. (Note though that this strategy cannot really be applied in other subjects such as math. Hudson 2020 points out some parallels and differences between linguistics and other school subjects, and I think this might be a new point in the context of his discussion.)

2) The authors correctly point out the limitations of their study, one of them being that

"there is no data to shed more light on students’ reasoning (for example from think aloud protocols), that could deepen our understanding of students’ understanding both within and outside of the intervention."

=> Could the authors point out ways to (systematically) implement such a qualitative component in future studies? It's always hard to analyze such protocols properly because researchers could basically pick out any aspect of it that might fit their findings best. How could such protocols be properly coded and analyzed according to the authors?

3)" For example, some students used the concept of role in an everyday sense rather than in a metalinguistic sense, indicating that they have not yet fully grasped the grammatical concept yet. Their use of the word role may therefore be seen as an indication of a limited understanding, although, encouragingly, it may also mean that students are on a conceptual journey, a part of which is to struggle with new terminology and concepts."

=> I was wondering whether this also holds for the term "grammar" itself. Our everyday concept of grammar is fundamentally different from the one in linguistics (prescriptive vs. descriptive, social norms vs. cognitive capacity etc.); see Döring (2020) for a recent paper on this. It would be nice if the authors could add this more general aspect to their final discussion.

References

Döring, Sandra (2020): Shaking students’ beliefs about grammar: Some thoughts on the academic education of future language teachers. In Andreas Trotzke & Tanja Kupisch (Hg.), Formal Linguistics and Language Education, 91-110. Cham: Springer.

Hudson, Richard (2020). Towards a pedagogical linguistics. Pedagogical Linguistics 1, 8–33.

6. PLOS authors have the option to publish the peer review history of their article (what does this mean?). If published, this will include your full peer review and any attached files.

Reviewer #1: No

Reviewer #2: No

---

## [Author Response · Author response to Decision Letter 0]

19 Nov 2021

Dear reviewers, please see the appended file for a response to your comments and suggestions. Thank you for your time - we feel that the manuscript has greatly improved as a result of your feedback.

---

## [Decision Letter · Decision Letter 1]

13 Jan 2022

Linguistic metaconcepts can improve grammatical understanding in L1 education: evidence from a Dutch quasi-experimental study

PONE-D-21-18569R1

Dear Dr. van Rijt,

We’re pleased to inform you that your manuscript has been judged scientifically suitable for publication and will be formally accepted for publication once it meets all outstanding technical requirements.

Kind regards,

Christina Manouilidou, Ph.D

Academic Editor

PLOS ONE

Additional Editor Comments (optional):

Reviewers' comments:

Reviewer's Responses to Questions

**Comments to the Author**

1. If the authors have adequately addressed your comments raised in a previous round of review and you feel that this manuscript is now acceptable for publication, you may indicate that here to bypass the “Comments to the Author” section, enter your conflict of interest statement in the “Confidential to Editor” section, and submit your "Accept" recommendation.

Reviewer #1: All comments have been addressed

Reviewer #2: All comments have been addressed

2. Is the manuscript technically sound, and do the data support the conclusions?

Reviewer #1: Yes

Reviewer #2: Yes

3. Has the statistical analysis been performed appropriately and rigorously? 

Reviewer #1: I Don't Know

Reviewer #2: Yes

4. Have the authors made all data underlying the findings in their manuscript fully available?

Reviewer #1: Yes

Reviewer #2: Yes

5. Is the manuscript presented in an intelligible fashion and written in standard English?

Reviewer #1: Yes

Reviewer #2: Yes

6. Review Comments to the Author

Reviewer #1: The authors have carefully considered the comments provided for the original manuscript and have changed it accordingly. Therefore, I have no further comments.

If the editor finds that the manuscript is suitable for publication, I am happy to support his/her decision.

Reviewer #2: All comments have been addressed.

7. PLOS authors have the option to publish the peer review history of their article (what does this mean?). If published, this will include your full peer review and any attached files.

Reviewer #1: No

Reviewer #2: No

---

## [Editor Report · Acceptance letter]

20 Jan 2022

PONE-D-21-18569R1 

Linguistic metaconcepts can improve grammatical understanding in L1 education
Evidence from a Dutch quasi-experimental study 

Dear Dr. van Rijt:

I'm pleased to inform you that your manuscript has been deemed suitable for publication in PLOS ONE. Congratulations! Your manuscript is now with our production department. 

Kind regards, 

on behalf of

Dr. Christina Manouilidou 

Academic Editor

PLOS ONE